# Ovarian Cancer Biomarkers in the COVID-19 Era

**DOI:** 10.3390/ijerph20115994

**Published:** 2023-05-29

**Authors:** Antonella Farina, Flavia Colaiacovo, Mariacarmela Gianfrate, Beatrice Pucci, Antonio Angeloni, Emanuela Anastasi

**Affiliations:** Department of Experimental Medicine, Sapienza University of Rome, 00161 Rome, Italy; antoeffe22@gmail.com (A.F.);

**Keywords:** ovarian cancer, SARS-CoV-2, tumor markers, HE4

## Abstract

Ovarian Cancer (OC) diagnosis is entrusted to CA125 and HE4. Since the latter has been found increased in COVID-19 patients, in this study, we aimed to evaluate the influence of SARS-CoV-2 infection on OC biomarkers. HE4 values above the cut-off were observed in 65% of OC patients and in 48% of SARS-CoV-2-positive patients (not oncologic patients), whereas CA125 values above the cut-off were observed in 71% of OC patients and in 11% of SARS-CoV-2 patients. Hence, by dividing the HE4 levels into quartiles, we can state that altered levels of HE4 in COVID-19 patients were mostly detectable in quartile I (151–300 pmol/L), while altered levels in OC patients were mostly clustered in quartile III (>600, pmol/L). In light of these observations, in order to better discriminate women with ovarian cancer versus those with COVID-19, we established a possible HE4 cut-off of 328 pmol/L by means of a ROC curve. These results demonstrate that the reliability of HE4 as a biomarker in ovarian cancer remains unchanged, despite COVID-19 interference; moreover, it is important for a proper diagnosis that whether the patient has a recent history of SARS-CoV-2 infection is determined.

## 1. Introduction

Ovarian cancer (OC), the most frequent gynecological malignancy affecting women worldwide, is the 5th leading cause of cancer-related death in women, where a late diagnosis, in more than 70% of cases, results in a low five-year survival rate [1,2,3]. The high lethality rate of OC is mainly attributable to several reasons: the lack of proper screening programs and the characteristic abdominal spreading with minimal clinical symptoms. The early detection of this tumor increases the five-year survival rate to 90% when neoplasia is confined to the ovary (Stage I), or 70% when it is confined to the pelvis (Stage II). However, most ovarian cancer is diagnosed at stages III (51%) and IV (29%), and thus the five-year survival rate is less than 30% [4].

Thus, the development of early detection methods for these diseases is clearly an urgent need. Novel biomarkers would also enable disease stratification, the monitoring of patients’ response to therapy, and the surveillance of post-surgical recurrence in women with a diagnosis of OC. To date, the diagnosis of OC is entrusted to two biomarkers: CA125, which is approved as a gold standard, and the more recent “secretory protein of the human epididymis 4” (HE4) [5,6,7], which is a secretory protein that is encoded by the gene whey acidic protein (WAP)-four disulfide core domain protein 2 that is localized on human chromosome 20q12–13.1 [8].

HE4 is among the most frequently upregulated genes in epithelial ovarian carcinomas based on gene expression profiles [9].

Several studies have reported that HE4, in addition to the ovarian epithelium, is also expressed in a wide spectrum of tissues, such as the lungs and salivary glands, thus functioning as a defense protein of the oral cavity and respiratory tract because of its anti-inflammatory activity and its involvement in natural immunity [10,11,12]. Interestingly, HE4 levels in COVID-19 patients have been found increased. In addition to HE4, the presence of circulating tumor markers such us SCC, CA19.9, CEA and CA125 have been detected in COVID-19 patients [11,13]. COVID-19 is a systemic illness caused by the novel SARS-CoV-2 (severe acute respiratory syndrome coronavirus 2) [14]. SARS-CoV-2 belongs to the Coronaviridae family, sharing common structural and biological properties with them. Four structural viral proteins (spike (S), envelope (E), membrane (M), and nucleocapsid (N)) have been identified as necessary for virion assembly and the suppression of the host immune response [15,16]. Primary infection occurs in the upper respiratory tract where the virus finds a suitable environment for replication [17]. In some patients, the pathology can evolve into acute respiratory distress (ARDS), pneumonia, multi-organ failure and, in the most severe cases, illness with death [18,19,20]. COVID-19 vaccines have emerged as an important strategy by which to induce robust humoral and cellular immunity and prevent adverse outcomes caused by SARS-CoV-2 infection [21,22,23]. The disease is generally correlated with the hyper-reaction of the immune system and the “cytokines storm”, which induces secondary tissue damage promoted by the over-secretion of active mediators and inflammatory factors. Cytokines and chemokines such as TNF-α, IL-6, IL-12 and IL-8 are profusely released [24,25,26,27]. Among inflammatory cytokines, IL-6 is dramatically increased in COVID-19 patients and it has been reported that more than half of admitted patients in the emergency unit showed elevated IL-6 levels [28].

Moreover, several studies have highlighted the role of this cytokine as an important marker of disease severity and a predictor of mortality [26,27].

IL-6 is a multifunctional cytokine that is promptly and transiently produced by T cells and macrophages in response to infections and tissue injuries, with a key role in host defense through the stimulation of acute phase responses, hematopoiesis, and immune reactions [29]. Inflammation promoted by viral infection induces an aberrant production of ferritin and IL-6 [30,31], increasing the mortality risk [32,33].

Serum ferritin, considered to be an “acute-phase reactant”, is able to mirror the degree of both the acute and chronic inflammatory reaction. However, investigations of ferritin values in COVID-19 patients have yielded unclear results. It is not evident whether it is a bystander or a real peculiarity of the disease [34]. Concerning this issue, retrospective studies have reported the minimal role of ferritin in predicting intensive care admission and the failure to predict mortality [35]. In contrast, another publication presented different scientific evidence, suggesting that ferritin assessment might be of crucial importance for the early identification of patients at a higher risk of an adverse outcome [36]. Taking int account that virus infection, inflammation, and tumors are tightly connected in a network [37], we aimed to evaluate the influence of SARS-CoV-2 infection on the tumor biomarkers closely associated with ovarian cancer. Considering this, HE4 and CA125 were evaluated in a population of SARS-CoV-2-positive women (with and without vaccination), compared to patients affected by ovarian cancer. Furthermore, HE4 levels were compared and correlated with IL-6 and ferritin, which are closely associated with the analytical parameters heavily modulated by COVID-19.

## 2. Materials and Methods

### 2.1. Patients

From April 2020 to March 2021, a prospective observational study was performed on women admitted to Policlinico Umberto I Hospital, Sapienza University of Rome, Italy. Patients with a prior history of a benign or malignant gynecological disease were excluded. For this study, we Enrolled 253 women, who were subdivided as follows:

Group (1) Thirty-seven unvaccinated SARS-CoV-2-swab-positive women (mean age 73.8 ± 11.6).

Group (2) Forty-eight SARS-CoV-2-swab-positive women that had previously received three COVID-19 vaccine inoculations (mean age 78.7 ± 9.8).

Group (3) Eighty-eight healthy women (equally distributed into vaccinated and unvaccinated) (mean age 60.5 ± 5.6) recruited from the biobank “Centro Trasfusionale” of the “Azienda Ospedaliera Policlinico Umberto I”, La Sapienza University of Rome.

Group (4) Eighty women affected by ovarian cancer (mean age 68.4 ± 9.0).

### 2.2. Serum Samples

Sera were acquired following a standard protocol. Briefly, samples were collected in a yellow-top vacutainer (Becton, Dickinson and Co., Plymouth, UK), clotted for 60–90 min and centrifuged for 10 min at 1300× *g*. The serum fractions obtained were then aliquoted in 1.5 mL Eppendorf tubes (Eppendorf S.r.l., Milan, Italy) and stored at −80 °C until analysis.

### 2.3. HE4 Assay

HE4 levels were determined using the HE4 non-competitive, indirect, two-step sandwich chemiluminescent immunoenzymatic (CLEIA) method conducted on a LUMIPULSE^®^ G1200 automated analyzer (Fujirebio Diagnostics, Seguin, TX, USA).

The two-incubation system involved the use of disposable cartridges containing capture microspheres coated with specific anti-HE4 monoclonal Abs (2H5 MAb) and the use of a second Ab labeled with alkaline phosphatase (ALP), directed towards the two epitopes of the antigen, forming a sandwich complex.

The immune complex was detected upon addition of a substrate solution containing 3-(2′-spiroadamantyl)-4-methoxy-4-(3″-phosphoryloxy)-phenyl-1,2-dioxetane (AMPPD), which was dephosphorylated by the catalysis of ALP indirectly conjugated to the particles.

Luminescence (at a maximum wavelength of 477 nm) was generated by the cleavage reaction of dephosphorylated AMPDD and detected using a luminometer.

The luminescent signal reflects the amount of HE4.

According to the manufacturer’s indications, normal values of HE4 were considered to be less than 150 pmol/L [38].

### 2.4. CA125 Assay

CA125 levels were determined using the CA125 non-competitive, indirect, two-step automated analyzer (Fujirebio, Tokyo, Japan).

The two-incubation system involved the use of disposable cartridges containing capture microspheres coated with specific anti-CA125 monoclonal Ab (OC125).

CA125 in the specimens specifically bound to anti-CA125 monoclonal antibodies on the particles, and antigen–antibody immune complexes were formed. A second alkaline phosphatase (ALP)-labeled anti-CA125 antibody monoclonal, directed towards the two epitopes of the antigen, formed a sandwich complex.

The immune complex was detected upon addition of a substrate solution containing 3-(2′-spiroadamantyl)-4-methoxy-4-(3″-phosphoryloxy)-phenyl-1,2-dioxetane (AMPPD), which was dephosphorylated by the catalysis of ALP indirectly conjugated to the particles.

Luminescence (at a maximum wavelength of 477 nm) was generated by the cleavage reaction of dephosphorylated AMPDD and detected using a luminometer.

The luminescent signal reflects the amount of CA125.

According to the manufacturer’s indications, normal values of CA125 were considered to be less than 35 U/mL [38].

### 2.5. IL-6 Assay

IL-6 levels were evaluated using a Electrochemiluminescent immunoassay (ECLIA) based on a one-step double-antigen sandwich principle on the ROCHE Cobas e601 analyzer system.

IL-6 determination involved a two-step quantitative double-antibody assay:-1st incubation: 18 µL of samples were incubated with a biotinylated monoclonal IL-6-specific antibody.-2nd incubation: After the addition of a monoclonal IL-6-specific antibody labeled with the Ruthenium (II) tris-bipyridyl complex and the streptavidin-coated microparticles, the antibodies formed a sandwich complex with the antigen of the sample.

The reaction mixture was aspirated into the measuring cell where the microparticles were magnetically captured onto the surface of the electrode. Unbound substances were then removed by washing with a buffer solution.

The application of a voltage to the electrode then induced the chemiluminescent reaction due to the decay of the excited state of the ruthenium complex by means of the reducing power of tripropylamine (tpa), which allows the emission of a photon with a wavelength of 620 nm. The chemiluminescent emission was measured by a photomultiplier and the signal was transmitted to a data acquisition module that converted the signal into a concentration expressed in pg/mL, by means of a two-point calibration master curve.

According to the manufacturer’s indications, normal values of IL-6 were considered to be less than 7 pg/mL [39].

### 2.6. Ferritin Assay

Ferritin levels were evaluated using a Electrochemiluminescent immunoassay (ECLIA) based on a one-step double-antigen sandwich principle on the ROCHE Cobas e601 analyzer system.

The determination of ferritin involved a quantitative one-step double-antibody assay:-1st incubation: 10 µL of sample, with a solution containing a biotinylated monoclonal ferritin-specific antibody, and a monoclonal ferritin-specific antibody labeled with the Ruthenium (II) tris-bipyridyl complex, formed a sandwich complex.-2nd incubation: After the addition of streptavidin-coated microparticles, the complex became bound to the solid phase via the interaction of biotin and streptavidin.

The reaction mixture was aspirated into the measuring cell, where the microparticles were magnetically captured onto the surface of the electrode. Unbound substances were then removed by washing with a buffer solution. The application of a voltage to the electrode then induced the chemiluminescent reaction due to the decay of the excited state of the ruthenium complex by means of the reducing power of tripropylamine (tpa), which enables the emission of a photon with a wavelength of 620 nm. 

The chemiluminescent emission was measured using a photomultiplier and the signal was transmitted to a data acquisition module that converted the signal into a concentration expressed in µg/L, by means of a two-point calibration master curve. 

According to the manufacturer’s indications, normal values were considered to be 30–400 µg/L for men and 13–150 µg/L for women [36,40].

### 2.7. Statistical Analysis

The statistical significance between the HE4 and CA125 values in the group of COVID-19-positive patients (Group 1 and Group 2) and in the group of healthy patients (Group 3) was evaluated using the Mann–Whitney U test. The correlation of the IL-6 and ferritin values with the HE4 values was evaluated using the Spearman R nonparametric test. The statistical significance between the values of HE4, which were divided into quartiles, for the patients with COVID-19 and patients with ovarian cancer was analyzed using the Chi-squared test. The statistical significance between the percentages of patients with COVID-19 (SARS-CoV-2 Group) and patients with ovarian cancer (Group 4) with CA125 > 35 U/mL was analyzed using the Chi-squared test. The ROC curve was created to calculate the cut-off (using Youden’s J Index) that best discriminated between patients with COVID-19 and patients with ovarian cancer and to obtain the sensitivity and specificity of the test. The data were presented as mean  ±  standard deviation (SD). A *p*-value < 0.05 was considered as statistically significant. For statistical analysis, GraphPad Prism Statistical Software version 7.00 and MedCalc Statistical Software version 14.8.1 were used.

## 3. Results

### 3.1. Evaluation of HE4 and CA125 Levels in SARS-CoV-2 Positive Patients (Unvaccinated and Vaccinated)

To verify whether COVID-19 could affect the levels of HE4 and CA125, both biomarkers were evaluated in Group 1 (unvaccinated patients) and Group 2 (vaccinated patients), respectively (see material and methods). HE4 values above the normal cut-off (>150 pmol/L) were observed in 49% of the Group 1 and in 48% of the Group 2 COVID-19 patients. A statistically significant increase in HE4 levels was found both in Group 1 and 2 compared to healthy women (Group 3) (respectively, 156.8 ± 78.43 vs. 53.42 ± 15.24 pmol/L, *p* value < 0.0001; 180.1 ± 86.81 vs. 53.42 ± 15.24 pmol/L, *p* value < 0.0001) (Figure 1a). CA125 values above the normal cut-off (>35 U/mL) were detected in 16% of Group 1 and in 6% of the Group 2 patients. No statistically significant difference was observed between the two groups, nor for Group 3 (respectively, 28.24 ± 42.43 vs. 13.46 ± 5.15 U/mL, 22.06 ± 28.35 vs. 13.46 ± 5.15 U/mL) (Figure 1b).

### 3.2. Correlations between HE4 vs. IL-6 and vs. Ferritin

In the two groups (1 and 2), the positivity rates for HE4 and CA125 were not statistically significant, so they were considered as a homogeneous group and called the SARS-CoV-2 Group. As mentioned above, following SARS-CoV-2 infection, the immune system can be dysregulated, leading to cytokine overproduction. Based on this observation, we investigated whether a correlation between HE4 levels and inflammatory biomarkers such as ferritin and IL-6, would be detectable in the SARS-CoV-2 group. Spearman’s correlation analysis showed a statistically significant correlation between HE4 and IL-6 levels (R = 0.3296, ** *p* value = 0.0025) (Figure 2a). Otherwise, no statistically significant correlation was observed between HE4 and ferritin (Figure 2b).

### 3.3. Comparison of HE4 and CA125 Pathological Values in Ovarian Cancer Patients and in SARS-CoV-2 Group

Taking into account the relevance of HE4 and CA125 in the diagnosis of ovarian cancer, we investigated the levels of these two tumor markers in the SARS-CoV-2 group compared to the ovarian cancer patients (OC) (Group 4). HE4 values above the cut-off (>150 pmol/L) were observed in 65% of Group 4 and in 48% of the SARS-CoV-2 group (Figure 3a). Meanwhile, CA125 values above the cut-off (>35 U/mL) were observed in 71% of Group 4 and in 11% of the SARS-CoV-2 group (*p* value < 0.0001) (Figure 3b).

To better understand the HE4 values observed, they were arbitrarily distributed in quartiles as follows: quartile I, values from 151 to 300 pmol/L; quartile II, values from 301 to 600 pmol/L; quartile III, values > 601 pmol/L. The HE4 distribution in quartiles is summarized in Table 1.

Patients with OC (Group 4) were distributed as follows: 19% in quartile I, 17% in quartile II and 64% in quartile III. The SARS-CoV-2 group patients were distributed as follows: 78% in quartile I, 22% in quartile II and 0% in quartile III. Interestingly, in this group, HE4 levels were mainly distributed in quartile I (Figure 4). 

Our results open up questions about the use of HE4 in the presence of COVID-19. Therefore, in order to identify a cut-off that is able to better classify OC patients, by using ROC analysis we have identified a threshold value of 328 pmol/L (AUC = 0.668; *p* value = 0.0003; sensitivity = 51.25%; specificity = 96.47%) (Figure 5).

## 4. Discussion

Since the onset of the SARS-CoV-2 pandemic, laboratory medicine has contributed to clarify the clinical characterization of COVID-19 patients [41]; to date, studies have suggested that several circulating biomarkers are deranged by COVID-19 infection, and some are considered predictors of a poor prognosis and mortality [42]. Vitamin k, associated with inflammation markers such as ferritin, PCR, D-dimer, IL-6, and the electrophoretic profile, has been widely applied to manage SARS-CoV-2-infected patients [39,43,44,45].

Since inflammation is often linked to alterations in tumor biomarkers, several studies have been conducted in order to evaluate the possible aberrant expression of neoplastic benchmarks in COVID-19 patients. In this regard, it has recently been demonstrated that the alteration of some biomarkers is closely related to gynecological cancer and several investigations have reported a significant variation of HE4 levels in SARS-CoV-2-infected patients [11,13]. The mucin CA125 is considered the gold standard tumor marker in OC patients, but its clinical use in the management of these neoplastic patients is limited because it is also frequently elevated in women with benign gynecologic disorders [46,47,48]. Therefore, there has been a growing interest in emergent ovarian cancer biomarkers; thus, the US Food and Drug Administration (FDA) has accepted human epididymal secretory protein 4 (HE4) as a good tumor marker for the clinical setting of OC patients [38].

In light of these observations, in this study, we aimed to verify the reliability of HE4 and CA125 in the diagnosis and follow-up of OC, also considering the possible interference of SARS-CoV-2 virus infection on biomarker levels.

In view of the large compliance with anti-SARS-CoV-2 vaccination protocol, we firstly compared HE4 and CA125 expression, in a cohort of unvaccinated and vaccinated COVID-19 patients. The increase in HE4 values between the two groups overlapped, so we considered them as a homogeneous cluster. In contrast, CA125 showed a slight value increase in both vaccinated and unvaccinated patients. Otherwise, Wei X et al. demonstrated the overproduction of CA125 in severe COVID-19 patients, probably due to the deep involvement of lung parenchyma inflammation, suggesting the implication of other pathophysiological events in CA125 tuning [13]. The different behavior of HE4 compared to CA125 demonstrated in this study could indicate the better clinical sensitivity of this biomarker, thus suggesting its potential application in the surveillance of asymptomatic forms of COVID-19.

Indeed, since SARS-CoV-2 infection can induce an inflammatory response with the hyperactivation of the immune system, we studied the possible correlations between inflammation markers such as IL-6 and ferritin and HE4 in COVID-19 patients. In agreement with previous studies, a positive correlation was observed between HE4 and IL-6 levels (R = 0.3296, ** *p* = 0.0025) [11]. One explanation for these results could be related to different events involved in the pathogenesis ofCOVID-19: severe lung inflammation, triggered by IL-6, and the involvement of HE4 in the respiratory defense mechanisms promoted by innate immunity.

It is known that ferritin mediates immune dysregulation and contributes to a cytokine storm via direct pro-inflammatory and immunosuppressive activities; furthermore, increased ferritin levels have been found in many viral infections [31,49]. In our study, we observed no correlation between ferritin and HE4, indicating that the documented ferritin increase in COVID-19 patients could be considered an independent factor in disease severity.

Hence, by dividing the HE4 levels into quartiles, we can state that altered levels of HE4 in COVID-19 patients were mainly detectable in quartile I, while those in OC patients were mainly clustered in quartile III. In light of these observations, to better discriminate between women with ovarian cancer and COVID-19 patients, we established a possible cut-off of 328 pmol/L by means of a ROC curve.

## 5. Conclusions

The reliability of HE4 as a biomarker in ovarian cancer remains unchanged despite the interference of COVID-19. Moreover it is important, for a proper diagnosis, to know whether the patient has contracted the virus or not.

Thus concluding, since the CA125 values are not influenced by the viral infection, it would be appropriate to support the evaluation of this biomarker also.

## Figures and Tables

**Figure 1 ijerph-20-05994-f001:**
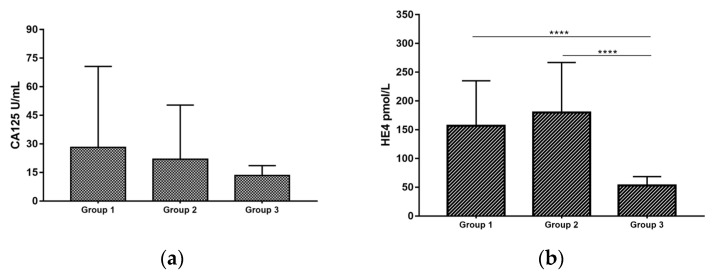
HE4 (**a**) and CA125 (**b**) levels in COVID-19 patients and Healthy donors. Histogram represents HE4 (**a**) and CA125 (**b**) percentage in Group 1, Group 2 and Group 3. The asterisks indicate significant differences between groups (**** *p* value < 0.0001).

**Figure 2 ijerph-20-05994-f002:**
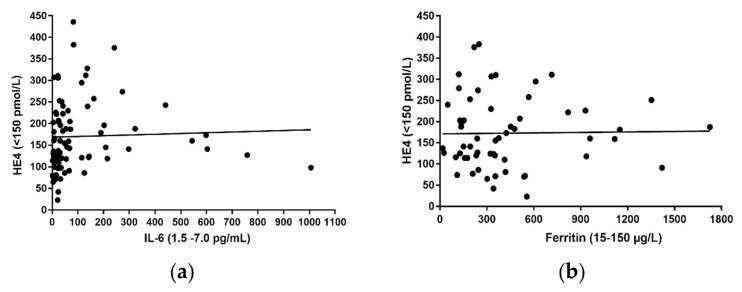
Correlation analysis of HE4 vs. IL-6 and vs. ferritin in SARS-CoV-2 group. (**a**) Spearman’s correlation analysis performed on HE4 levels (expressed in pmol/L) compared to IL-6 values (expressed in pg/mL) (R = 0.3296). (**b**) Spearman’s correlations analysis applied to HE4 values (expressed in pmol/L) with respect to ferritin values (expressed in µg/L).

**Figure 3 ijerph-20-05994-f003:**
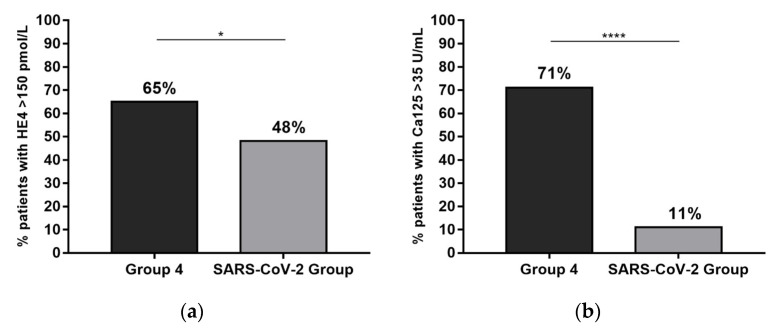
HE4 and CA125 percentages above the cut-off (HE4 > 150 pmol/L, CA125 > 35 U/mL) in Group 4 patients and in the SARS-CoV-2 group. (**a**) Histogram of the percentages of the values of HE4 > 150 pmol/L in Group 4 and the SARS-CoV-2 group; (**b**) histogram of the percentages of the values of CA125 > 35 U/mL in the same two groups of patients. The asterisks indicate significant differences between groups (* *p* value = 0.03, **** *p* value <0.0001).

**Figure 4 ijerph-20-05994-f004:**
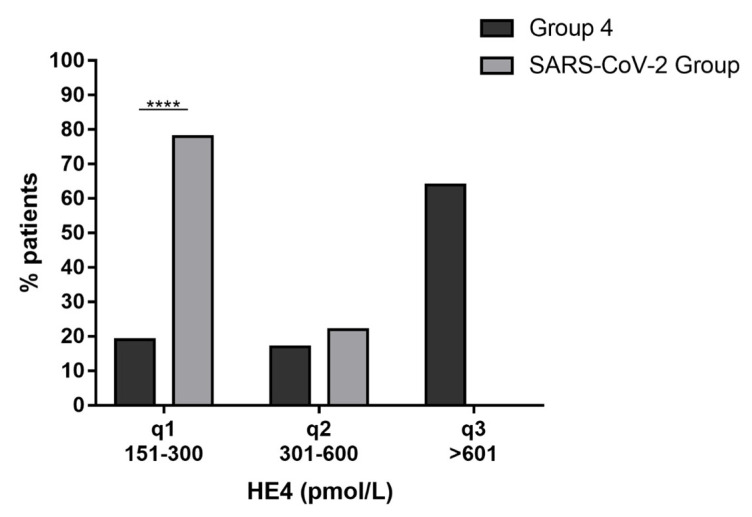
Evaluation of HE4 levels in OC vs COVID-19. Statistical analysis showing the percentage distribution of HE4 levels in the selected population. The asterisks indicate a significant difference between groups (**** *p* < 0.0001).

**Figure 5 ijerph-20-05994-f005:**
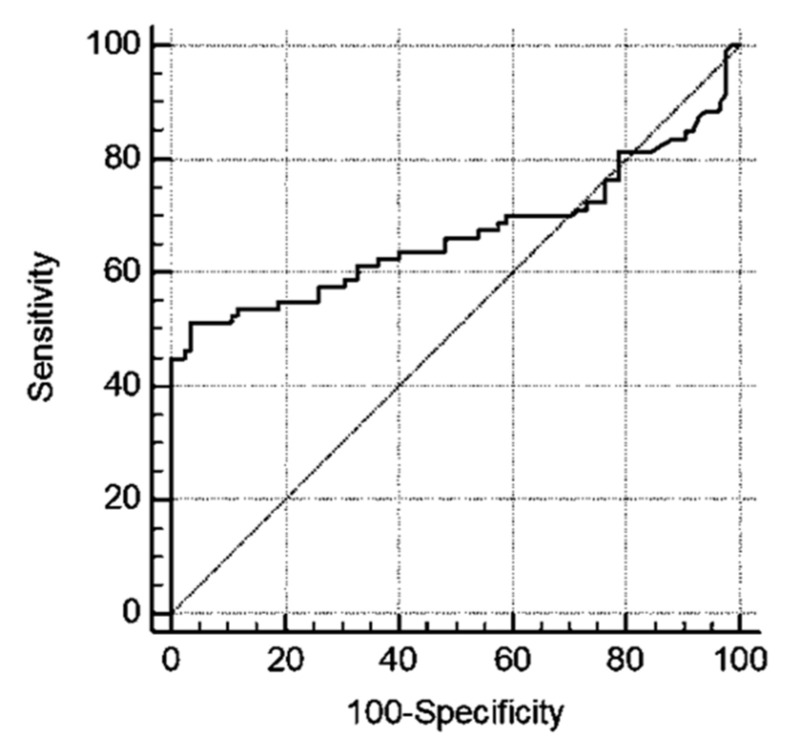
ROC regression analysis of HE4 in OC vs. COVID-19 patients. AUC = 0.668; *p* value = 0.0003; sensitivity = 51.25%; specificity = 96.47%.

**Table 1 ijerph-20-05994-t001:** Subdivision into quartiles of positive HE4 values observed in the selected women’s population.

	% q1	% q2	% q3
HE4 (pmol/L)	151–300	301–600	>601
Group 4	19	17	64
SARS-CoV-2 group	78	22	0
Group 3	0	0	0

## Data Availability

Not applicable.

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
