# Peer review of "Ovarian Cancer Biomarkers in the COVID-19 Era"

_ijerph, 2023, doi:10.3390/ijerph20115994_

Round 1

Reviewer 1 Report

Authors presented two biomarkers from OC and presented their possible application in COVID patients. The retrospective study finally proposed a cut off value of HE4 for predictions. I have the following questions to the authors.

1.      The data shows that vaccinated patients have higher HE4, did you have a subgroup between vaccinated COVID positive and vaccinated healthy individual?

2.      From group 4, did you have any history of COVID vaccination status? How does the marker vary between OC with vaccine and without vaccine?

3.      How does this quartile distribution method differentiate OC patients based on the marker? While is it clear that there is difference in the pmol /L of HE4 what other reasoning could we give after the quartile distribution?

4.      What are the limitations of the current study?

5.      What are the other confounding factors in the study design?

Author Response

Authors presented two biomarkers from OC and presented their possible application in COVID patients. The retrospective study finally proposed a cut off value of HE4 for predictions. I have the following questions to the authors.

  1. The data shows that vaccinated patients have higher HE4, did you have a subgroup between vaccinated COVID positive and vaccinated healthy individual?

Thank you for the question. Our results clearly show HE4 increase both in vaccinated and unvaccinated COVID-19 patients (thus were grouped); whereas healthy women did not display HE4 values above the cut-off. Thus, suggesting that SARS-CoV-2 natural infection induces HE4 alteration.

  1. From group 4, did you have any history of COVID vaccination status? How does the marker vary between OC with vaccine and without vaccine?

Thank you for the question. To rule out SARS-CoV-2 interference on HE4 expression, we intentionally selected from our bio bank serum samples collected from OC patients diagnosed before the pandemic.

  1. How does this quartile distribution method differentiate OC patients based on the marker? While is it clear that there is difference in the pmol /L of HE4 what other reasoning could we give after the quartile distribution?

The arbitrary quartile division allowed us to understand the differences in HE4 levels and their distribution between Ovarian Cancer and COVID-19 patients. As reported in the manuscript, while OC patients are mainly grouped in the III quartile (HE4 >601pmol/L), COVID-19 patients were clustered in the I quartile (151-300 pmol/L). Based on this observation we can speculate that COVID-19 patients have a lower increase in HE4 value respect to OC.

  1. What are the limitations of the current study?

       Since we have no data regarding the duration of HE4 alteration promoted by viral infection, further studies are needed to better clarify this issue.

  1. What are the other confounding factors in the study design?

      To this regard we believe that SARS-CoV-2 viral variants can exert different biological interferences reflecting on HE4 levels. To date unfortunately we can’t discriminate the impact of viral variants on tumor markers modulation. This point could be a suggestion for further research projects.

Reviewer 2 Report

The authors present results of the influence of the SARS-CoV-2 virus on well-known biomarkers for ovarian cancer, HEK4, and CA125.
The results showed that the HEK4 protein is overexpressed in COVID-positive patients and CA125 is unchanged compared to healthy patients. According to the results, to distinguish between COVID-positive and OC patients, they propose a cut-off level of HEK4 to be 328 pmol/L. They conclude that these results demonstrate that the reliability of HE4 as a biomarker in ovarian cancer remains unchanged despite COVID-19 interference.

The research is well-explained and written.

Author Response

Thank you for your revision

Reviewer 3 Report

The study by Farina et al., attempts to draw a correlation between overlapping biomarkers observed in ovarian cancer vs SARS-CoV2 infection such as CA125 and HE4. It is an interesting study that adds value to determine the appropriate biomarker for accurate ovarian cancer diagnosis.

1.       Ovarian cancer is “one of the most frequent gynecologic cancers” not the most frequent as mentioned. It is the deadliest amongst gynecologic cancers. Please change line 24.

2.       Make the methods more concise, only describing what is specific to this study. Cite sources when appropriate to avoid detailed descriptions of the protocols.

3.       “Histogram” misspelled in several instances.

4.       In Figure 1: plot healthy donors (Group 3) first and Groups 2 and 3 subsequently as this is a standard practice and would make it easy for readers to interpret. Label groups as what they stand for rather than labelling them as Group1/2/3.

5.       Figure 3: Stay consistent with labelling throughout the manuscript. Change “Group 4” to what it stands for.

6.       Figure 3 and 4: Denote error bars on histograms.

Author Response

  1. Ovarian cancer is “one of the most frequent gynecologic cancers” not the most frequent as mentioned. It is the deadliest amongst gynecologic cancers. Please change line 24.

Ok, it has been changed.

  1. Make the methods more concise, only describing what is specific to this study. Cite sources when appropriate to avoid detailed descriptions of the protocols.

Ok, it has been changed.

  1. “Histogram” misspelled in several instances.

Ok, it has been corrected.

  1. In Figure 1: plot healthy donors (Group 3) first and Groups 2 and 3 subsequently as this is a standard practice and would make it easy for readers to interpret. Label groups as what they stand for rather than labelling them as Group1/2/3.

Ok, it has been corrected.

  1. Figure 3: Stay consistent with labelling throughout the manuscript. Change “Group 4” to what it stands for.

Ok, it has been changed.

  1. Figure 3 and 4: Denote error bars on histograms.

Histograms in figures 3 and 4 are only used to graphically represent total percentages, therefore there is no standard error, moreover GraphPad program does not allow to add error bars in this case.

Reviewer 4 Report

COVID19 has swept the world and it has a great effect on academic field, which makes itself a spotlight for many research scientists. The authors evaluate the effects of COVID19 infection on the diagnostic power of well-established OC biomarkers (CA125 and HE4). CA125 and HE4 are free from the interference of COVID19 in terms of diagnosis for OC.

Minor issues:

1, in figure 2, the correlations between IL6/Ferritin and HE4 is very weak even though they are significant at cutoff value.

2, you can also add some introductions about correlation between cancer and COVID19 in your introduction section.

Author Response

1, in figure 2, the correlations between IL6/Ferritin and HE4 is very weak even though they are significant at cutoff value.

Thank you for pointing out this observation, we discussed this result in Pag 8 line 274 (without Track Changes): “One explanation of these results could be related to different events attending the COVID-19 pathogenesis: severe lung inflammation, triggered by IL-6, and the involvement of HE4 in the respiratory defense mechanisms promoted by innate immunity.”

2, you can also add some introductions about correlation between cancer and COVID19 in your introduction section.

Ok, It has been added see pag 2 line 80 (without Track Changes)

Reviewer 5 Report

The article entitled  Ovarian Cancer biomarkers in COVID-19 era is a document of interesting subject matter.

The idea of the work is good and the topic is important. However, the reported work  requires improvement and revisions.
1. Your abstract should clearly state the essence of the problem you are addressing, what you did and what you found and recommend. That will help a prospective reader of the abstract to decide if they wish to read the entire article.
2. paper must provide a comprehensive critical review of recent developments in a specific area or theme. It is expected to have an extensive literature review followed by an in-depth and critical analysis of the state of the art, and identify challenges for future research.

3. The authors should do the analysis the conclusion section must clearly establish a strong correlation with the proposed topic.
4. The conclusion section can be refined better. Please indicates if your objectives were reached, in what your work is novel and confirms or not, previous findings. Also, ‎‎some perspectives generally arise from your investigations and must be indicated here. ‎‎

5. The interpretation of the experimental results should be significantly improved. In other words, pay attention on the interpretation of the experimental results. Only their presentation is not enough for a scientific paper.

6. The abstract and conclusion parts must be more informative by including more mathematical findings.

Author Response

  1. Your abstract should clearly state the essence of the problem you are addressing, what you did and what you found and recommend. That will help a prospective reader of the abstract to decide if they wish to read the entire article.

Unfortunately, the number of words required limits the accurate description of the paper. The paper contains many concepts and data, but we have been limited to describe the most significant concepts. Therefore, the abstract has been partially changed according to your suggestions.

  1. paper must provide a comprehensive critical review of recent developments in a specific area or theme. It is expected to have an extensive literature review followed by an in-depth and critical analysis of the state of the art and identify challenges for future research.

We consider very important this suggestion that could be a starting point for an original review. This is a preliminary study published as a research article. Studies on this topic are few and unfortunately not to allow an extensive literature review followed by an in-depth and critical analysis of the state of the art.

  1. The authors should do the analysis the conclusion section must clearly establish a strong correlation with the proposed topic.

We believe that this point it has been largely analyzed in the discussion line from 256 to 283 (without Track Changes)?

  1. The conclusion section can be refined better. Please indicates if ‎your objectives were‎reached, in what your work is novel and confirms or not, previous findings. Also, ‎‎some perspectives generally arise from your investigations and must be indicated here.‎‎

We better refined conclusion section, and for what concerning previous findings we commented them in discussion session.

  1. The interpretation of the experimental results should be significantly improved. In other words, pay attention on the interpretation of the experimental results. Only their presentation is not enough for a scientific paper.

Thank you, this is a preliminary report, and the results should be significantly improved in a new project.

  1. The abstract and conclusion parts must be more informative by including more mathematical findings.

Unfortunately, the number of words required limits the accurate description of the paper. The paper contains many concepts and data, but we have been limited to describe the most significant concepts. Therefore, the statistically analysis has been added.